# The role of CO$_2$ in the genesis of Dabie-type porphyry molybdenum deposits

Zi-Qi Jiang[1], Lin-Bo Shang [1]✉, A. E. Williams-Jones[2], Xin-Song Wang [1]✉, Li Zhang[3], Huai-Wei Ni[3], Rui-Zhong Hu[1] & Xian-Wu Bi[1]

Porphyry-type molybdenum deposits, many of which are in China, supply most of the World's molybdenum. Of particular importance are the molybdenum deposits located in the Qinling-Dabie region that are responsible for more than half of China's molybdenum production. A feature that distinguishes this suite of deposits from the better-known Climax and Endako subtypes of porphyry molybdenum deposits is their formation from CO$_2$-rich magmatic-hydrothermal fluids. The role of CO$_2$, if any, in the transport of molybdenum by these fluids, however, is poorly understood. We conducted experiments on the partitioning of molybdenum between H$_2$O-CO$_2$, H$_2$O-NaCl, and H$_2$O-NaCl-CO$_2$ fluids and a felsic melt at 850 °C and 100 and 200 MPa. Here we show that the exsolution of separate (immiscible) brine and vapor leads to the very high brine $D_{Mo}$ values needed for efficient extraction of Mo from the magmas forming Dabie-type porphyry molybdenum deposits.

Porphyry molybdenum deposits have been subdivided into three main types, Climax, Endako, and Dabie[1–4]. The third of these types, which occurs in China, accounts for more than 30% of the World's molybdenum resource[5]. Unlike the subduction-related Climax-type and Endako-type porphyry molybdenum deposits, the Dabie-type porphyry molybdenum deposits developed in a post-collisional extensional environment[6,7]. In addition, the ore-forming fluids for the Dabie-type molybdenum deposits are characterized by high proportions of CO$_2$, and primary fluid inclusion assemblages containing coexisting CO$_2$ vapor and brine inclusions are common in the main ore stage, notably in the giant Shapinggou porphyry Mo deposits of the Dabie orogen[4,6–12]. This, the fact that the homogenization temperatures of the primary brine inclusions commonly exceed 500 °C (ref. 8) and the lack of evidence for the entrapment of supercritical fluids, suggest that the corresponding magmas exsolved separate CO$_2$ and high salinity aqueous fluids. However, the role of CO$_2$, if any, in partitioning molybdenum into the exsolving phase(s) and the formation of Dabie-type porphyry molybdenum deposits has not been investigated.

Numerous experimental studies have investigated the partitioning of Mo between fluids and melt as a function of fluid salinity, $f$O$_2$, melt composition, and, more recently, as a function of these

parameters and $f$S$_2$ and pressure[13–19]. At low salinity, the value of $D_{Mo}$ is generally less than 10, whereas for high salinity, values as high as 135 have been reported[16]. In addition, the experiments conducted by Tattitch and Blundy[16] and Zhao et al.[18], led to the conclusion that D$_{Mo}$ increases linearly with increasing fluid salinity. The effect of $f$O$_2$ is subordinate to that of salinity, with higher $D_{Mo}$ values associated with higher values of $f$O$_2$[16]. The same is true of $f$S$_2$ and pressure, increases of which also lead to higher $D_{Mo}$ values[17]. Molybdenum partitions more strongly into the aqueous phase exsolving from peralkaline magmas than that exsolving from peraluminous magmas[17,19]. This is because molybdenum dissolves in the magma as molybdate moieties, e.g., Na$_2$MoO$_4$ and/or K$_2$MoO$_4$ in peralkaline melts[20], analogs of which, e.g., NaHMoO$_4^0$, are very stable in aqueous fluids, leading to a strong preference of Mo for the fluid[21].

Two experimental studies have investigated the effect of CO$_2$ on the solubility of molybdenum (molybdenite) in aqueous fluids[22,23]. The studies were carried out at temperatures up to 450 °C and 600 °C, respectively. They found that the solubility of molybdenum either remained constant[22] or decreased slightly with increasing $X_{CO_2}$ above 0.1 (ref. 23).

[1]State Key Laboratory of Ore Deposits Geochemistry, Institute of Geochemistry, Chinese Academy of Sciences, Guiyang, China. [2]Department of Earth and Planetary Sciences, McGill University, Montreal, QC, Canada. [3]CAS Key Laboratory of Crust-Mantle Materials and Environments, School of Earth and Space Sciences, University of Science and Technology of China, Hefei, China. ✉e-mail: shanglinbo@vip.gyig.ac.cn; wangxinsong@mail.gyig.ac.cn

The partitioning of molybdenum between $CO_2$-bearing aqueous fluids and felsic melts, necessary for understanding the genesis of Dabie-type molybdenum deposits, is not well constrained. Such a study would be important for a fuller understanding of the genesis of Dabie-type molybdenum deposits and the role of $CO_2$ in this genesis. In this paper, we report the results of experiments designed to investigate the partitioning of molybdenum between $CO_2$-bearing and -free aqueous fluids and felsic melt at temperature and pressure conditions relevant to the formation of Dabie-type porphyry molybdenum deposits. Our experiments elucidate the role of $CO_2$ in this formation. Specifically, we address the issue of whether the presence of $CO_2$ affects the partitioning of molybdenum between a felsic magma and an exsolving hydrothermal fluid and, in turn, whether this affects the genesis of Collisional- or Dabie-type porphyry molybdenum deposits.

## Results & Discussion

### The results of the partitioning experiments

Our experiments exhibited significantly different behavior in the series of experiments. In the $H_2O$-$CO_2$ experiments, which employed the leaching method to determine the Mo concentration of the fluid, the $D_{Mo}^{fluid/melt}$ ($C_{Mo}^{fluid}/C_{Mo}^{melt}$, in which $C_{Mo}^{fluid}$ and $C_{Mo}^{melt}$ represent the concentration of Mo in the equilibrated fluids and quenched glasses, respectively) was $0.2 \pm 0.02$ (2σ) at $X_{CO2} = 0.1$, $0.34 \pm 0.04$ and $0.4 \pm 0.04$ (2σ) at $X_{CO2} = 0.15$ and $0.25 \pm 0.02$ (2σ) at $X_{CO2} = 0.2$ (Supplementary Table 5) (where $X_{CO2}$ represents the mole ratio of $CO_2/(CO_2 + H_2O)$ in the experiments). Thus, values of the partition coefficient for Mo do not correlate with $X_{CO2}$ in NaCl-free systems involving $H_2O$-$CO_2$ fluids (see below).

In the $H_2O$-NaCl experiments, all the fluid inclusions had the same vapor-liquid ratio at room temperature (Fig. 1a). Based on the equation of state for the $H_2O$-NaCl system at the experimental conditions (850 °C, 200 MPa), the fluid lies in the single-phase region. Values of $D_{Mo}^{fluid/melt}$, calculated using the concentration of Mo in the leaching solution, ranged from $0.51 \pm 0.28$ (2σ) to $21.3 \pm 10.3$ (2σ), whereas $D_{Mo}^{fluid/melt}$ values, calculated using the concentration of Mo in fluid inclusions, ranged from $0.66 \pm 0.33$ (2σ) to $25.7 \pm 14.0$ (2σ). The two sets of $D_{Mo}^{fluid/melt}$ values, however, are indistinguishable within the analytical uncertainty of ~ 20% (Supplementary Fig. 2b). In general, the $D_{Mo}^{fluid/melt}$ value increases with increasing salinity (Fig. 2b). At low salinity (≤20 wt.%), the $D_{Mo}^{fluid/melt}$ values (calculated using the fluid inclusion data) vary from $0.66 \pm 0.33$ to $3.18 \pm 1.41$ (2σ), whereas at higher salinity, and especially at a salinity above that of halite saturation, the partition coefficient of molybdenum increases sharply, from $6.08 \pm 2.48$ (2σ) at a NaCl concentration of 21.2 wt.% to $25.7 \pm 14.0$ (2σ) at a NaCl concentration of 44.3 wt.% (Supplementary Table 5).

In the $H_2O$-NaCl-$CO_2$ experiments, brine and vapor were trapped as separate (immiscible) phases as shown by the presence of vapor and brine inclusions, as well as heterogeneously (variable proportions of vapor and brine), in quartz from the experiments for this system (Fig. 1b, c, d). This indicates that the melt equilibrated with separate brine and vapor phases in this system at the experimental conditions (850 °C, 200 MPa). As a result of this phase separation, the salinity of the brine increased from that of the homogeneous system (~ 7 wt.%) to 56 wt.% NaCl at a $X_{CO2}$ of 0.1 and 62 wt.% NaCl at a $X_{CO2}$ of 0.3. The $D_{Mo}^{brine/melt}$ value increased from $81 \pm 41.1$ to $179 \pm 69.7$ (2σ) as the salinity increased from 56 wt.% NaCl ($X_{CO2} = 0.1$) to 62 wt.% NaCl ($X_{CO2} = 0.3$) (Fig. 2c and Supplementary Table 5).

### The influence of salinity on $D_{Mo}$

The results of this study clearly show that the partition coefficient of Mo between hydrothermal fluids and felsic magmas ($D_{Mo}^{fluid/melt}$) increases with the salinity of the fluid and that this increase is exponential with salinity (Fig. 2b, c). Thus, hypersaline fluids are very efficient in leaching Mo from coexisting felsic melts. Several studies have reported trends of $D_{Mo}^{fluid/melt}$ values with salinity similar to that

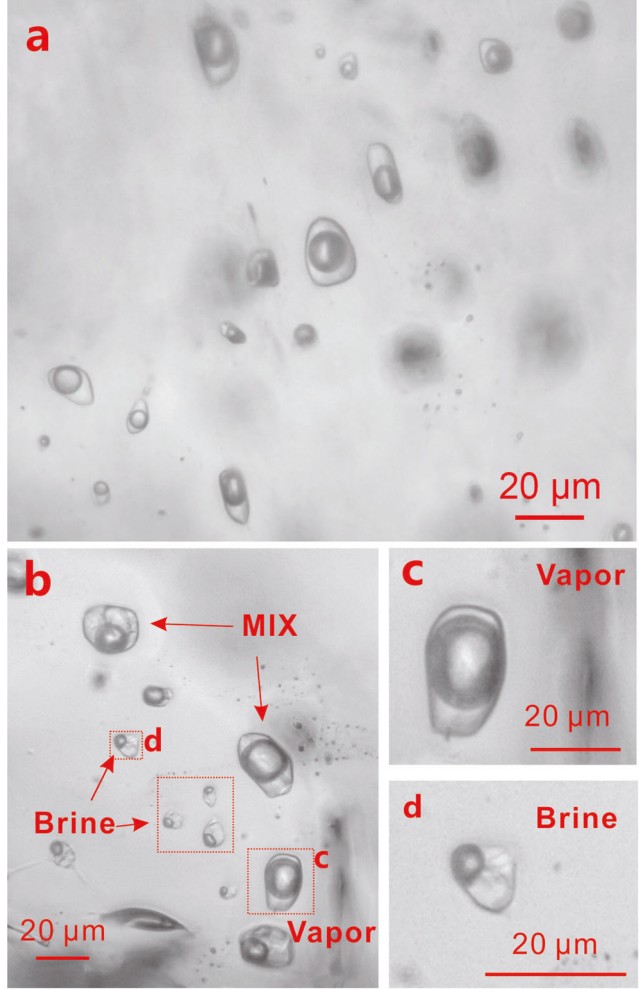

**Fig. 1 | Photomicrographs of fluid inclusions trapped in quartz during experiments at 850 °C and 200 MPa. a** Supercritical $H_2O$-NaCl fluid inclusions from experiment Q-006. **b** Coexisting low-density vapor inclusions, high-density brine inclusions, and heterogeneously trapped vapor and brine inclusions in the $H_2O$-NaCl-$CO_2$ system from experiment Q-016. **c, d** are enlargements of one of the vapor and brine inclusions in (**b**), respectively.

reported here[15–18]. In all cases the $D_{Mo}^{fluid/melt}$ value for the high salinity fluids was reported to be ≥100 times higher than that for low salinity fluids. These data are illustrated in Fig. 3 and are consistent with experimental and theoretical findings that Mo is transported dominantly by the species $NaHMoO_4^0$ (ref. 21) or, at very high temperatures, by the species $MoO_2(OH)Cl$ (ref. 24), for a wide range of salinity. We, therefore, conclude that any process that leads to the formation of a hypersaline brine in equilibrium with a fertile felsic magma will lead to an ore fluid that is enriched in Mo and capable of the deposition of this metal in concentrations sufficient to produce economic porphyry Mo deposits.

### The role of $CO_2$ on $D_{Mo}$

The results of our study show that $CO_2$ is not complex with Mo (Fig. 2a) and thereby increases the ability of aqueous carbonic fluids to extract Mo from magma. On the contrary, our experiments with $CO_2$-$H_2O$ fluids show that an increase in the $CO_2$ content of the system does not affect the Mo content of the fluid or the fluid-melt partition coefficient for Mo, which were both very low (Supplementary Table 5). The same conclusion was reached by Li et al.[23], However, as we show below, $CO_2$ plays a very important, albeit indirect, role in the leaching of Mo from felsic melts and the formation of Mo-rich ore fluids.

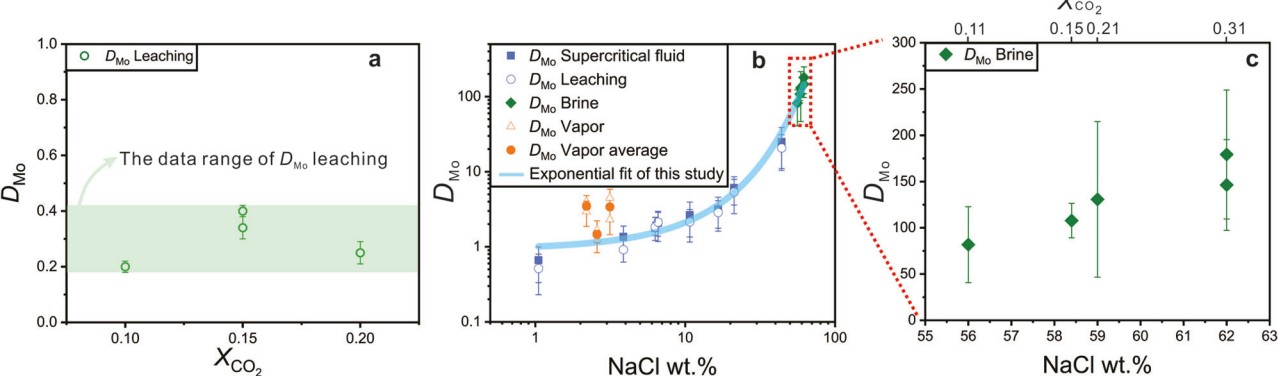

**Fig. 2 | The $D_{Mo}$ vs. $X_{CO2}$ and salinity of fluid. a** $D_{Mo}^{fluid/melt}$ vs. $X_{CO2}$ for the $H_2O$-$CO_2$ experiments. The $D_{Mo}^{fluid/melt}$ values were calculated from the leaching solution analyzed by ICP-MS and quenched melt analyzed by LA-ICP-MS. **b** $D_{Mo}^{fluid/melt}$ vs. salinity in the $H_2O$-NaCl and $H_2O$-NaCl-$CO_2$ experiments. The solid squares represent the $D_{Mo}^{fluid/melt}$ values calculated from the concentrations of Mo in fluid inclusions and the quenched melt of the $H_2O$-NaCl experiments. The blank circles represent the $D_{Mo}^{fluid/melt}$ values calculated from the leaching solution and quenched melt of the $H_2O$-NaCl experiments, and the solid diamonds represent the partition coefficients for Mo between brine and melt for the $H_2O$-NaCl-$CO_2$ experiments. Because of the large uncertainty in the estimates of the salinity of the vapor fluid inclusions in the $H_2O$-NaCl-$CO_2$ experiments, only maximum and minimum values of $D_{Mo}^{vapor/melt}$ are shown. The solid cycle represents the average value of the above value. **c** An enlargement of the rectangular box in (**b**) showing the values of $D_{Mo}^{fluid/melt}$ for the high salinity experiments in the $H_2O$-NaCl-$CO_2$ system. The corresponding $X_{CO2}$ is also indicated.

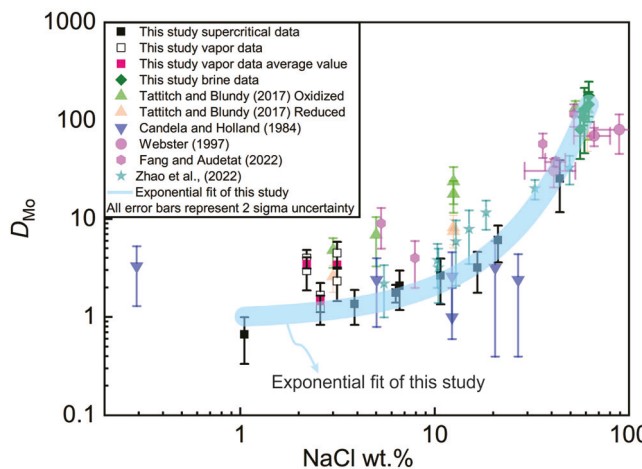

**Fig. 3 | Compilation of $D_{Mo}^{fluid/melt}$ between fluids and melts from experiments with different temperatures and pressure, and 2 sigmas standard deviation of the measurements was illustrated.** The experimental data was collected from Candela and Holland[13], Webster[15], Tattitch, and Blundy[16], Fang and Audetat[17], Zhan et al.[18], and this study. The values increase exponentially with salinity. The values reported by Tattitch and Blundy[16], and Fang and Audetat[17] are consistently higher than those of the current study. A possible reason for this is the presence of sulfur, which is interpreted by the authors of these papers to increase the solubility of Mo in the fluids relative to that of Mo in sulfur-free experiments.

In experiments with $H_2O$-NaCl-$CO_2$ fluids, the fluid separated into immiscible brine and vapor at a temperature of 850 °C and 200 MPa, provided that the $X_{CO2}$ was greater than 0.1. At lower $X_{CO2}$ and in the system $H_2O$-NaCl system at the same temperature and pressure there is a single supercritical fluid (Fig. 4a), consistent with the observation of Li et al.[25], that the addition of $CO_2$ expands the two-phase region of the $H_2O$-NaCl fluid system and promotes fluid immiscibility.

The maximum value of $D_{Mo}^{brine/melt}$ determined in this study for the $H_2O$-NaCl-$CO_2$ system is ~100 times higher than that determined for an $H_2O$-NaCl fluid having the same bulk NaCl content. The reason for this is the very strong partitioning of NaCl into the liquid and the resultingly low NaCl content of the coexisting vapor. Consequently, the liquid in the $H_2O$-NaCl-$CO_2$ system will have a much higher NaCl content than the supercritical fluid in an $H_2O$-NaCl system with the same NaCl content as that of the combined fluids in the $H_2O$-NaCl-$CO_2$ system. As the $D_{Mo}^{fluid/melt}$ value of the brine increased sharply with NaCl content at the higher end of the range of NaCl contents considered, it follows that the $D_{Mo}^{fluid/melt}$ of the brine in the $H_2O$-NaCl-$CO_2$ system will reach a value many times higher than that for the corresponding $H_2O$-NaCl system.

An important feature of the behavior of $CO_2$ in magma is that it promotes the saturation of the magma with fluid[26–28]. Moreover, it also promotes the exsolution of the fluid as separate $CO_2$-rich vapor and NaCl-rich liquid because, as mentioned above, the immiscibility region in the system $H_2O$-NaCl-$CO_2$ increases with increasing $X_{CO2}$ (ref. 27). Thus, at the temperature-pressure conditions of the emplacement of magmas forming Dabie-type porphyry Mo deposits (850 °C and 200 MPa), the magma will exsolve an ore-forming hydrothermal fluid as separate (immiscible) vapor and brine phases, even if the bulk fluid contains as little as 10 mole % $CO_2$.

In summary, the principal findings of our study are: 1) that $D_{Mo}^{fluid/melt}$ values increase with increasing NaCl content of the fluid, particularly at the upper end of the range of NaCl contents considered; and 2) that, for the temperature-pressure conditions of our experiments (similar to those for the emplacement of the magmas forming Dabie-type porphyry Mo deposits) and $X_{CO2} > 0.1$, the melt coexists with vapor and a hypersaline brine with a higher NaCl content than that of the supercritical fluid in the $H_2O$-NaCl system having the same bulk NaCl content. The main implication of these findings is that liquid-vapor phase separation induced by the presence of $CO_2$ during the exsolution of $H_2O$-NaCl fluids from magmas may be the key to the efficient extraction of Mo from a fertile magma and the formation of an economic Dabie-type porphyry Mo deposit.

## Implications for Dabie-type porphyry molybdenum ore formation

As mentioned in the introduction to this paper, Dabie-type porphyry molybdenum deposits, which postdated the Qinling-Dabie continent-continent collision, are the main Chinese source of Mo, accounting for over half of the resource of this metal in China and one-third of the resource globally. In contrast to the hydrothermal fluids that formed Climax- and Endako-type porphyry molybdenum deposits, the hydrothermal fluids responsible for the formation of Dabie-type porphyry Mo deposits are of higher temperature, higher salinity, and $CO_2$-

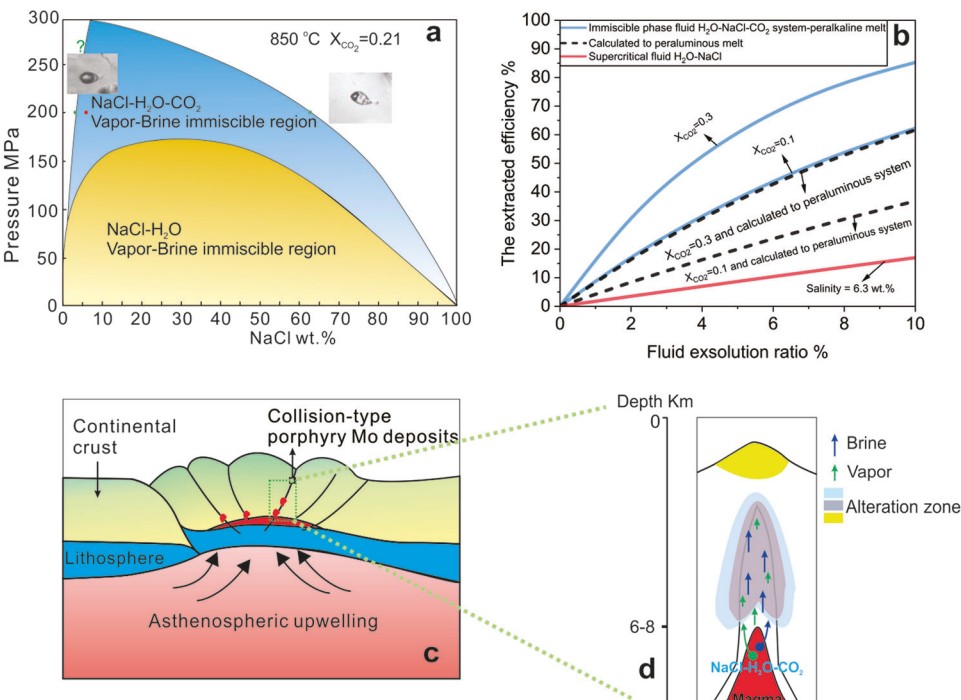

**Fig. 4 | The phase diagrams of H₂O-NaCl and H₂O-NaCl-CO₂, extracted efficiency of Mo from exsolved fluid and schematic diagrams of the formation of Dabie-type porphyry molybdenum deposits. a** A pressure-salinity diagram showing phase relationships in the systems $H_2O$-NaCl, and $H_2O$-NaCl-$CO_2$ calculated from Driesner and Heinrich[43] and Duan et al.[44], respectively. **b** The extraction efficiency ratio of molybdenum, defined as the mass of Mo in the exsolved fluid phase divided by the mass of Mo in the initial magma, versus the ratio of exsolved fluid to magma. The $D_{Mo}^{fluid/melt}$ values obtained from our experiments were used to calculate the extraction efficiency. The blue line represents the extracted efficiency of Mo from initial magma when $X_{CO2}$ = 0.1, and 0.3, respectively. The Black dashed line represents the extracted efficiency calculated for the peraluminous system. The red line represents the extracted efficiency in $H_2O$-NaCl supercritical fluid. Details of the calculation are provided in the supplemental materials. **c** The schematic diagram illustrated the post-collisional extensional environment[45] of Dabie-type or Collision-type porphyry molybdenum deposits. **d** The diagram shows the fluid evolution mole for Dabie-type porphyry molybdenum deposits, featuring immiscible vapor and brine $CO_2$ bearing ore-forming fluids that directly exsolve from felsic magma.

rich[4]. Indeed, $CO_2$-bearing vapor inclusions and coexisting halite-bearing fluid inclusions are ubiquitous in Dabie-type porphyry Mo deposits, e.g., the Shapinggou deposit, the largest Mo deposit in Asia and the Tangjiaping deposit, another large Mo deposit[8,10].

Like the fluid inclusions trapped during the formation of Dabie-type Mo deposits, those trapped during our experiments with the $H_2O$-NaCl-$CO_2$ system comprise assemblages of coexisting $CO_2$-rich and halite-bearing liquid-vapor inclusions. Accordingly, we propose that the hydrothermal fluids forming Dabie-type porphyry Mo deposits exsolved directly from the magma as separate $CO_2$-rich vapor and hypersaline brine phases. This contrasts with the cases of the Climax- and Endako-type deposits, for which the common occurrence of intermediate density fluid inclusions below the deposits and separate brine and vapor inclusions at the level of the deposits provide evidence that the corresponding magmas exsolved a single supercritical fluid that subsequently underwent phase separation[29–31].

To evaluate the efficiency with which a brine in the system $H_2O$-NaCl-$CO_2$ can extract Mo from a magma, we calculated the extraction efficiency using the $D_{Mo}^{fluid/melt}$ values determined from our experiments (details of the calculation are provided in the Supplementary S5). The results of this calculation show that when 10 % of a two-phase fluid with a brine/vapor ($CO_2$) ratio of 0.1 and a bulk salinity of ~ 7 wt.% is exsolved directly from the magma, it will extract 62.3% and 85.3% of the Mo from the magma at a $X_{CO2}$ of 0.1 and 0.3, respectively. In contrast, a supercritical fluid with a NaCl content of 6.3 wt.%, will only extract 17.0 % of the Mo from the magma. Thus, the efficiency of extraction of Mo by a brine in the $H_2O$-NaCl-$CO_2$ system containing a separate vapor is 3.7 ~ 5 times higher than that of supercritical fluid in the $H_2O$-NaCl system with the same bulk salinity as the $H_2O$-NaCl-$CO_2$

system. In addition, if the extraction efficiency is calculated using the ASI of the peraluminous magmas associated with porphyry molybdenum deposits (the $D_{Mo}$ for peraluminous melts is approximately half that for peralkaline melt[17,19]), the extraction of Mo is still 2 – 3 times higher than that with supercritical $H_2O$-NaCl fluids (Fig. 4b).

Based on the results of our experiments and the extraction efficiency of Mo from magmas, we envisage the following scenario for the genesis of Dabie-type porphyry molybdenum deposits. Ore formation begins with the production of a fertile magma from the partial melting of crustal material due to upwelling of the asthenosphere[4,32] (Fig. 4c). This magma is enriched in $CO_2$ because of the elevated carbonate content of the crust, in NaCl because of an evaporite component or sediments[9,32] and in Mo because of the incompatible behavior of Mo during the generation of the crust from the mantle. On emplacement at a depth corresponding to ~ 200 MPa, this magma undergoes $CO_2$-induced fluid exsolution and, because of the high $CO_2$ content, exsolves a $CO_2$-rich vapor and a separate hypersaline liquid, which sequesters the Mo. Finally, owing to the overpressures created by the fluid exsolution, fractures are created into which the liquid is dispersed to form a quartz-molybdenite stockwork (Fig. 4d). We propose that the scenario envisaged here satisfactorily explains the formation of Dabie-type Mo deposits and note that the exsolution of a two-phase fluid is supported by the boiling or effervescent fluid inclusion assemblages that characterize most Dabie-type porphyry Mo deposits[8,10,11].

To conclude, the results of our experiments show that the $H_2O$-NaCl-$CO_2$ fluids in equilibrium with felsic magmas at the conditions of emplacement of Dabie-type deposits occur as separate $CO_2$-rich vapors and hypersaline brines, that the unusually high salinity of the brines results in extremely high fluid/melt partition coefficients for

Mo, and that this enables extremely efficient extraction of Mo from the magma. Accordingly, we propose that $CO_2$ plays an important, if not controlling role in the formation of Dabie-type porphyry Mo deposits by inducing fluid-exsolution at greater depths than would be possible for Climax- or Endako-type deposits and by ensuring that two fluids are exsolved. The result is a hypersaline brine that has the high NaCl content needed to extract most of the Mo from the magma and facilitate the formation of a Collisional- or Dabie-type porphyry Mo deposit.

## Methods

To explore the role of $CO_2$-rich ore-forming fluids in the formation of Dabie-type porphyry molybdenum deposits, experiments were conducted to determine the partitioning of molybdenum between fluids in the systems $H_2O$-$CO_2$, $H_2O$-NaCl, and $H_2O$-NaCl-$CO_2$ and felsic melt. The $H_2O$-$CO_2$ series of experiments was carried out at 850 °C and 100 MPa, and involved the use of the quenched fluid leaching method of Keppler and Wyllie[14] (1991) for determination of the Mo concentration in the fluid, whereas in the $H_2O$-NaCl and $H_2O$-NaCl-$CO_2$ experiments, which were conducted at 850 °C and 200 MPa, the fluid was trapped in synthetic fluid inclusions for subsequent analysis. After the experiments, the quenched leaching fluid was analyzed by ICP-MS and the fluid inclusions trapped in the quartz cylinder and quenched melt were analyzed by LA-ICP-MS. A detailed description of the experimental setup and analytical methods are as follows.

### Experiments

The experimental starting glass was synthesized using pre-determined masses of $Na_2CO_3$, $KHCO_3$, $Al_2O_3$, and $SiO_2$ that were finely ground in an agate mortar under acetone. Molybdenum was added as $MoO_3$. The resulting mixture was placed in a platinum crucible and heated at 1000 °C for 12 h in a silicon-molybdenum rod furnace, after which it was heated to 1350 °C for 2 h and quenched. The quenched glass was ground and heated at 1350 °C for a further 2 h and quenched again. These three steps were repeated three times to ensure elemental homogeneity. A piece of the resulting glass was polished and analyzed for its composition using LA-ICP-MS (Supplementary Table 1). The remaining glass was ground to 200 mesh and used as the initial melt phase for the experiments. The fluid composition used in the experiments is reported in Supplementary Table 2. In the $H_2O$-$CO_2$ series and the $H_2O$-NaCl-$CO_2$ series of experiments, oxalic acid dihydrate was used to produce $CO_2$. In the $H_2O$-NaCl and $H_2O$-NaCl-$CO_2$ series of experiments, solid NaCl was added to maintain the salinity of the system and in the high $X_{CO_2}$ experiments to counter the effect of the decomposition of oxalic acid dihydrate to produce water.

All the experiments were conducted in cold-seal vessels at the University of Science and Technology of China in Hefei. For the $H_2O$-$CO_2$ series of experiments, 50 mg of the melt and predetermined masses of $H_2O$ and oxalic acid dihydrate were sealed in a gold capsule (supplementary Fig. 1a). The outer diameter of the gold capsule is 5 mm, the inner diameter is 4.6 mm, and the length is 20 mm. The experimental conditions were 850 °C and 100 MPa. After the experiments, the leaching solution was analyzed to determine the Mo concentration in the fluid.

To determine the state of the $H_2O$-NaCl and $H_2O$-NaCl-$CO_2$ fluids under the equilibrium experimental conditions, we used a quartz cylinder to trap the fluid. As the diffusion coefficient of Mo in the melt is very low, we first synthesized fluid inclusions containing ~1000 ppm Rb by trapping them in a fractured quartz cylinder at the experimental conditions. The quartz cylinder containing these fluid inclusions was then fractured in situ after equilibrium (7 days) and kept at the experimental temperature and pressure for another 5 days to trap the fluid present after the attainment of equilibrium[33] (Supplementary Fig. 1b). Approximately 1000 ppm Cs was added to all the experimental solutions to check that the fluid inclusions analyzed by LA-ICP-MS were those trapped after the "in-situ" fracturing.

For the $H_2O$-NaCl series experiments, 50 mg of the melt, 50 mg of the starting solution, and the quartz cylinder containing the fluid inclusions were sealed together in a gold capsule. The outer diameter of the gold capsule is 5 mm, the inner diameter is 4.6 mm, and the length is 20 mm. Oxygen fugacity was not controlled during the experiments, but since the autoclave body is a nickel-based alloy, the oxygen fugacity was likely close to that of the NNO buffer[34]. The experiments were conducted at 850 °C and 200 MPa for 7 days, after which the sample holder was transferred to the cold end of the furnace and held for 10 s. In this way, the fluid inclusions containing Rb were destroyed by thermal shock (Supplementary Fig. 1b). The sample holder was then returned to the high-temperature end of the furnace to continue the reaction for 5 days, thereby ensuring complete capture of the equilibrated fluid. After the experiments were completed, a slow quench was performed, pulling the sample chamber from the high-temperature end to the cold end of the furnace over an interval of 30 s. This ensured that the fluid inclusions would not decrepitate and that the melt would be quenched to glass. The steps for the $H_2O$-NaCl-$CO_2$ series of experiments were the same as for the $H_2O$-NaCl experiments, except that a pre-determined mass of solid NaCl was added to the capsules to buffer the salinity.

At the end of the experiment, the capsule was removed from the pressure vessel and weighed. If the mass difference before and after an experiment was less than 0.0005 g, the experiment was considered successful.

### Sample treatment after experiments

Two methods were used to measure the concentration of Mo in the equilibrium fluids after the experiments.

The first method, which was applied to the quenched fluid from experiments in the $H_2O$-$CO_2$ and $H_2O$-NaCl systems, involved leaching the charges with distilled water using a modification of the method of Keppler and Wyllie[14]. The cleaned capsules were placed in liquid nitrogen for a few seconds and, after being frozen, were cut open with scissors. Each of the opened capsules was placed in a separate Teflon beaker, to which pure water was added, and the beaker was warmed on an electric heating plate for several hours at a constant temperature of 80 °C. The leaching solutions (~40 ml from each experiment) were then transferred to tubes of known mass and centrifuged. The above steps were repeated three times. The Mo concentration in the leaching solution was analyzed using ICP-MS. This method could not be used to determine the composition of the quenched fluid from experiments in the $H_2O$-NaCl-$CO_2$ system because it contained separate brine and vapor phases at the experimental conditions.

The second method of determining the concentration of Mo in the reacted fluid involved trapping the fluid as inclusions in quartz at the conditions of the experiment. As mentioned above, a quartz cylinder was introduced into the gold capsules for this purpose. The method was applied to experiments in the $H_2O$-NaCl, and $H_2O$-NaCl-$CO_2$ systems. The quartz cylinder was prepared by heating and rapidly quenching it in $H_2O$ to create fractures and subsequently heating it to remove any $H_2O$ (ref. 33). After an experiment, a slice of the quartz cylinder was doubly polished, and the fluid inclusions (Fig. 1) were analyzed by LA-ICP-MS to obtain the Mo concentration in the fluid[35]. The salinity of the fluid inclusions, which was analyzed micro-thermometrically, was used as an internal standard.

The results of the two analytical methods used to determine the concentration of Mo in the equilibrated fluids of the $H_2O$-NaCl experiments were in good agreement, both in terms of the Mo concentration in the equilibrium solution and the final calculated $D_{Mo}$ values. This shows that both methods yield reliable results and provide confidence that the Mo concentration of the fluid in the $H_2O$-$CO_2$

experiments was reliably determined and could be used to calculate the $D_{Mo}$ values for this series of experiments (Supplementary Fig. 2).

The quenched glass was analyzed by LA-ICP-MS for its major element composition and Mo concentration.

## Analysis

We analyzed the fluid inclusions obtained from the $H_2O$-NaCl and the $H_2O$-NaCl-$CO_2$ experiments microthermometrically to provide an internal Na standard for subsequent LA-ICP-MS analyses of individual fluid inclusions. The microthermometric measurements for the fluid inclusions were conducted on a Linkam THMSG 600 programmable heating-freezing stage mounted on a Leica microscope at the State Key Laboratory of Ore Deposit Geochemistry (SKLODG), Institute of Geochemistry, Chinese Academy of Science (IGCAS) in Guiyang. Liquid nitrogen was used to freeze the fluid inclusions. The equipment permits the measurement of phase changes from $-196\,^\circ C$ to $600\,^\circ C$ and was calibrated using the microthermometric behavior of synthetic fluid inclusions of known composition. The results are reported in Table S3. The salinity of the fluid inclusions trapped during the $H_2O$-NaCl experiments was determined from the melting temperature of the ice except for experiment KD-003 (ref. [36]). In the latter experiment (KD-003), halite was present in the brine inclusions at ambient temperature and consequently, the melting temperature of halite was used to estimate the salinity[37]. The salinity of the vapor inclusions in the system $H_2O$-NaCl-$CO_2$, which did not contain a halite crystal, was determined from the decomposition temperature of the clathrate. Because of the large uncertainty associated with observing clathrate decomposition, only the salinity of the fluids in experiments Q-011, Q-016, and KD-005 could be reliably determined. Moreover, we only retained data for those inclusions that returned low and very similar salinity values, i.e., we excluded all outliers with high salinity that might represent the entrapment of brine with the vapor. We did not report the average vapor salinity, instead, we reported the range of salinity. The halite dissolution temperature was used to obtain the salinity of the other brine fluid inclusions from the $H_2O$-NaCl-$CO_2$ series of experiments. The brine salinity was calculated using data for the $H_2O$-NaCl system due to a lack of reliable data for $CO_2$-bearing systems. The salinity determined is assumed to have been accompanied by a 1% error based on the findings of Schmidt and Bodnar[38] and Nagasaki and Hayashi[39].

## LA-ICP-MS analysis of fluid inclusions and quenched glasses

The composition of individual inclusions was analyzed using a 193 nm excimer laser system from Coherent, and an Agilent 7900 inductively coupled plasma mass spectrometer at SKLODG. During the ablation, helium was used as the carrier gas, which was mixed with 5 ml/min of nitrogen before entering into the ICP-MS to improve the sensitivity of the analyzed elements. During an analysis, the laser operating frequency was 11 Hz, the energy was 11 J/cm$^2$, and the spot size was between 24 and 32 µm depending on the size and depth of the fluid inclusions. The spot size for the external standard, NIST 610, was 32 µm. Masses of $^{23}$Na, $^{29}$Si, $^{27}$Al, $^{39}$K, $^{85}$Rb, and $^{133}$Cs, were analyzed with dwell times of 20 ms; a dwell time of 50 ms was used to estimate the mass of $^{95}$Mo. Before an analysis, NIST 610 was used to optimize the performance of the ICP-MS and ensure that the instrument reached its highest sensitivity and ionization efficiency (U/Th≈1), that the oxide yields (ThO/Th < 0.3%) were as small as possible, and that the background values were low[35]. The compositions were determined using NIST 610 as the external standard and the microthermometrically determined salinity as the internal standard[40]. A representative LA-ICP-MS signal of a fluid inclusion analysis is shown in Supplementary Fig. 3. Data processing (including the selection of sample and blank signals, correction of instrument sensitivity drift, and elemental content calculation) was done using the SILLS software[41]. A representative analytical signal is illustrated in Supplementary Fig. 3; in this brine fluid inclusion from an $H_2O$-NaCl-$CO_2$ experiment, the concentration of Mo reached 15,424 ppm. We are not implying, however, that such a high concentration might be realized in nature as the Mo concentrations of the magmas are likely to be much lower than those of the melt employed in our experiments.

The analysis of the quenched glasses was also carried out at SKLODG using the equipment described above. Approximately ten points were randomly analyzed from the rim to the center of fragments of the quenched glass using the LA-ICP-MS to obtain the Mo and major element concentrations of the felsic melt. This made it possible to assess the homogeneity of the Mo concentration in the quenched glass and thereby determine whether the experiments reached equilibrium. Helium was used as the carrier gas for the laser ablation, and each sampling period consisted of ~ 20 s of blank signal and ~ 50 s of sample signal. Element contents were calibrated against multiple-reference materials (NIST 610, BCR-2G, BIR-1G, and BHVO-2G) without applying internal standardization. Off-line selection and integration of background and analyte signals, time-drift correction, and quantitative calibration were performed using ICPMSDataCal software[42].

## Reporting summary

Further information on research design is available in the Nature Portfolio Reporting Summary linked to this article.

## Data availability

The authors declare that all data generated or analyzed during this study are included in this published article (and its supplementary information files). The data also deposited in the repository of Figshare (https://figshare.com/s/5488cf08a83cb36695a6). Source data are provided with this paper.

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

## Acknowledgements

This work was supported financially by the National Key Research and Development Program of China (2022YFC2903303), the National Natural Science Foundation of China (42303074, 41673067), the West Light

Foundation of Chinese Academy of Sciences (grant xbzg-zdsys-202108) and the State Key Laboratory of Ore Deposit Geochemistry (grant 202201). We thank Hai-Hao Guo, Ting-Guang Lan, and Yan-Wen Tang for their technical assistance with synthetic fluid inclusions and LA-ICP-MS analysis.

## Author contributions

Author contributions: Z.Q.J, L.B.S. and X.S.W. conceived the study and designed the experiments. Z.Q.J. wrote the original manuscript, A. E. W.-J., X.S.W., L.B.S., H.W.N., R.Z.H. and X.W.B. reviewed/edited the manuscript and made many constructive suggestions that improved its quality considerably. L.Z. and H.W.N. helped with the experiments. R.Z.H., L.B.S., X.W.B. and X.S.W. supervised the project and acquired the funding.

## Competing interests

The authors declare no competing interests.
