## [Peer Review File · Nature Communications]

REVIEWER COMMENTS

Reviewer #1 (Remarks to the Author):

The paper by Jiang et al., is the first paper focusing on the role of CO₂ in the formation of the most important subtype of porphyry Mo deposit, i.e. the Dabie-type. They used two different methods to constrain the partitioning of Mo between fluid and coexisting granitic melt. The work is novel and of great importance, and the results seem reliable.

The only thing I concern comes from the melt they used. As reported in Table S1, there is no CaO was introduced, and the A/NK values were much lower than the ore-causative granitic rocks in natural Collision-type systems (normally larger than 1). Why? To be more confident, an initial melt similar to the natural granitic porphyry would be a better choice. I recommend publication after minor revision.

Line 127: DMofluid/melt, not DMofluid/melt

Lines 184-185: these deposits, mainly formed during Late Jurassic to Early Cretaceous, postdated the collision, and thus should be post-collisional. For details, see Chen YJ et al., 2017, *Ore Geol. Rev.*, or Li N et al., 2018, *Earth-Sci. Rev.*

223-225: the reason for Mo not only because of the incompatible behavior of Mo during the creation of the crust from the mantle, but also multiple pulses of further enrichment during the evolution of a continental collision. See Chen YJ et al., *Geology and Geochemistry of Molybdenum Deposits in The Qinling Orogen, P R China*. Springer.

Fig. S2: Between the background and the ablation of quartz and the fluid inclusion it hosted, there are several peaks for Si, Na, Al and Cs. Why? What does that mean?

Reviewer #2 (Remarks to the Author):

This paper reports new experimental data that clearly show the impact of CO₂ on fluid/melt partitioning of Mo. The data are applied to understanding the evolution of Dabie-type porphyry-Mo deposits, which are of significant importance as a Mo resource. This paper is a major advance in our understanding of Dabie-type systems. The data appear to be robust, the experiments well done and the interpretations are rooted firmly in the new data. I recommend accepting this paper with only a few minor changes.

L38-41: Regarding "Dabie-type porphyry molybdenum deposits developed in a post-collisional tensional environment", can you add some more details for the tectonic environment? Crustal thickness? Does extension result in "special" magmas? Is high CO₂ a feature of mantle melting? Or assimilation of carbonate? Is Cu and/or Au mineralization associated with these deposits?

L59-61: Can you add a sentence as to why fluid/melt partitioning of Mo is different for the different magma compositions?

L74: peralkaline melt? If yes, replace "felsic" with "peralkaline".

167: delete "the" after "system"

Responses to reviewers

Ref: NCOMMS-24-00298A

Reviewer #1

1. The paper by Jiang et al., is the first paper focusing on the role of CO₂ in the formation of the most important subtype of porphyry Mo deposit, i.e. the Dabie-type. They used two different methods to constrain the partitioning of Mo between fluid and coexisting granitic melt. The work is novel and of great importance, and the results seems reliable.

The only thing I concern comes from the melt they used. As reported in Table S1, there is no CaO was introduced, and the A/NK values were much lower than the ore-causative granitic rocks in natural Collision-type systems (normally larger than 1). Why? To be more confident, an initial melt similar to the natural granitic porphyry would be a better choice. I recommend publication after minor revision.

Response: The main purpose of the experiments discussed in this manuscript was to investigate the effect of CO₂ on the partitioning of Mo between aqueous fluids and felsic melts. To keep the system simple, we chose not to add CaO to the starting glass. The reason for this is that Ca²⁺ (the dominant form of Ca in the silicate melt) is a network modifying cation in silicate glasses (Stebbins, 2016). Thus, as shown in our previous study (Jiang et al., 2021), if present in significant proportions it would have made interpretation of the effect of the essential modifying cations, Na⁺ and K⁺, on the partitioning of Mo between fluid and felsic melts (D_{Mo}) much more difficult.

The reason that we chose to use a A/NK composition in our experiments that was lower than those of the granites associated with porphyry Mo deposits is that we were concerned that use of a higher A/NK would lead to D_{Mo} values, particularly for intermediate salinity fluids, that would be too small to determine reliably (Jiang et al., 2021; Fang and Audetat, 2022; Zhao et al., 2022). To compensate for the effect of this lower, peralkaline A/NK, which results in D_{Mo} values roughly twice as large as for a peraluminous A/NK (Jiang et al., 2021; Fang and Audetat, 2022), we adjusted the bulk

partition coefficient of Mo for the H₂O-NaCl-CO₂ system by reducing it by a factor of two and recalculated the corresponding extraction efficiency. As illustrated in **Figure 4b**, the recalculated extraction efficiency of the immiscible H₂O-NaCl-CO₂ fluid is still 2 to 3 times higher than that of the supercritical fluid of the H₂O-NaCl system with the same salinity. We, therefore, conclude that the increase in $D_{\text{Mo}}^{\text{bulk}}$ caused by using a peralkaline melt instead of a peraluminous melt does not change our conclusion that immiscibility in the system H₂O-NaCl-CO₂ leads to a much greater extraction of Mo from the melt than by a single phase fluid in the system H₂O-NaCl.

2. Line 127: $D_{\text{Mo}}^{\text{fluid/melt}}$, not $D^{\text{Mofluid/melt}}$

Response: Corrected.

3. Lines 184-185: *these deposits, mainly formed during Late Jurassic to Early Cretaceous, postdated the collision, and thus should be post-collisional. For details, see Chen YJ et al., 2017, Ore Geol. Rev., or Li N et al., 2018, Earth-Sci. Rev.*

Response: We have revised the phrase “during collision” to “post-collisional”.

4. 223-225: *the reason for Mo not only because of the incompatible behavior of Mo during the creation of the crust from the mantle, but also multiple pulses of further enrichment during the evolution of a continental collision. See Chen YJ et al., Geology and Geochemistry of Molybdenum Deposits in The Qinling Orogen, P R China. Springer.*

Response: Although there may have been be multiple episodes of enrichment in the evolution of the Qinling orogen, the purpose of the text at this point in the manuscript is to explain why the magma is enriched in CO₂, NaCl and Mo.

5. Fig. S2: *Between the background and the ablation of quartz and the fluid inclusion it hosted, there are several peaks for Si, Na, Al and Cs. Why? What does that mean?*

Response: These peaks are an artifact of the laser ablation technique that was employed to increase the success rate of fluid inclusion ablation. A small pre-ablation

small spot size (stepwise increase of beam size) was applied to avoid unexpected decrepitation of the quartz-hosted (Gunther et al., 1998). These peaks were per-ablation signals obtained before fluid inclusion ablation (Supplementary Figure. 2).

Reviewer #2

1. L38-41: *Regarding "Dabie-type porphyry molybdenum deposits developed in a post-collisional tensional environment", can you add some more details for the tectonic environment? Crustal thickness? Does extension result in "special" magmas? Is high CO₂ a feature of mantle melting? Or assimilation of carbonate? Is Cu and/or Au mineralization associated with these deposits?*

Response: As requested, additional details on the tectonic environment and its relationship to magmatism have been provided in the revised manuscript (see supplemental materials). The Dabie-type porphyry molybdenum deposits developed in a post-collisional extensional environment. The tectonic change from collisional compression to extension was associated with collapse, delamination, and thinning of over-thickened orogenic crust and lithosphere, accommodating large-scale magmatism and mineralization (Chen et al., 2017; Li et al., 2018). As a result, the granitic rocks related to ore-formation are metaluminous to peraluminous, and high-K calc-alkaline to shoshonitic. In addition, the Sr/Y ratios of these crust-derived granites decrease with their age from ~140 Ma to <127 Ma, reflecting thinning from over-thickened crust to crust of normal thickness (<35 km) in the area (Chen et al., 2017; Li et al., 2018). The carbon isotope compositions of fluid inclusions are in the range from -2.3 ‰ to +2.7 ‰ (Wang et al., 2014) suggesting that the CO₂ of the magmas originated from a mixture of recycled carbonate rocks and mantle. Rare Au/Cu mineralization is associated with the Mo mineralization.

2. L59-61: *Can you add a sentence as to why fluid/melt partitioning of Mo is different for the different magma compositions?*

Response: A sentence of explanation has been added. Molybdenum partitions more strongly into the aqueous phase exsolving from peralkaline magmas than that exsolving from peraluminous magmas. This is because molybdenum dissolves in the magma as molybdate moieties, e.g., Na_2MoO_4 and/or K_2MoO_4 in peralkaline melts, analogues of which, e.g., NaHMoO_4^0 , are very stable in aqueous fluids (Shang et al., 2020), leading to a strong preference of Mo for the fluid.

3. L74: *peralkaline melt? If yes, replace "felsic" with "peralkaline"*

Response: It was a “felsic melt”. In this part of the manuscript, we the term “felsic melt” is appropriate to describe the magma responsible for the formation of porphyry molybdenum deposits.

4. 167: *delete "the" after "system"*

Response: Corrected.

Chen, Y.-J., Wang, P., Li, N., Yang, Y.-F., and Pirajno, F., 2017, The collision-type porphyry Mo deposits in Dabie Shan, China: *Ore Geology Reviews*, v. 81, p. 405–430, doi:10.1016/j.oregeorev.2016.03.025.

Fang, J., and Audétat, A., 2022, The effects of pressure, $f\text{O}_2$, $f\text{S}_2$ and melt composition on the fluid–melt partitioning of Mo: Implications for the Mo-mineralization potential of upper crustal granitic magmas: *Geochimica et Cosmochimica Acta*, v. 336, p. 1–14, doi:10.1016/j.gca.2022.08.016.

Gunther, D., Audetat, A., Frischknecht, R., and Heinrich, C.A., 1998, Quantitative analysis of major, minor and trace elements in fluid inclusions using laser ablation inductively coupled plasma mass spectrometry: *Journal of Analytical Atomic Spectrometry*, v. 13, p. 263–270, doi:10.1039/a707372k.

Jiang, Z., Shang, L., Guo, H., Wang, X., Chen, C., and Zhou, Y., 2021, An experimental investigation into the partition of Mo between aqueous fluids and felsic melts: Implications for the genesis of porphyry Mo ore deposits: *Ore Geology Reviews*, v. 134, p. 104144, doi:10.1016/j.oregeorev.2021.104144.

Li, N., Chen, Y.-J., Santosh, M., and Pirajno, F., 2018, Late Mesozoic granitoids in the

- Qinling Orogen, Central China, and tectonic significance: *Earth-Science Reviews*, v. 182, p. 141–173, doi:10.1016/j.earscirev.2018.05.004.
- Shang, L., Williams-Jones, A.E., Wang, X., Timofeev, A., Hu, R., and Bi, X., 2020, An Experimental Study of the Solubility and Speciation of MoO₃(s) in Hydrothermal Fluids at Temperatures up to 350°C: *Economic Geology*, v. 115, p. 661–669, doi:10.5382/econgeo.4715.
- Stebbins, J.F., 2016, Glass structure, melt structure, and dynamics: Some concepts for petrology: *American Mineralogist*, v. 101, p. 753–768, doi:10.2138/am-2016-5386.
- Wang, P., Chen, Y.-J., Fu, B., Yang, Y.-F., Mi, M., and Li, Z.-L., 2014, Fluid inclusion and H–O–C isotope geochemistry of the Yaochong porphyry Mo deposit in Dabie Shan, China: a case study of porphyry systems in continental collision orogens: *International Journal of Earth Sciences*, v. 103, p. 777–797, doi:10.1007/s00531-013-0982-5.
- Zhao, P., Zajacz, Z., Tsay, A., Chu, X., Cheng, Q., and Yuan, S., 2022, The partitioning behavior of Mo during magmatic fluid exsolution and its implications for Mo mineralization: *Geochimica et Cosmochimica Acta*, p. 115–126, doi:10.1016/j.gca.2022.10.020.

REVIEWERS' COMMENTS

Reviewer #1 (Remarks to the Author):

The authors have made a revision according to comments/suggestions. I'd like to recommend accept for publication.